# Emerging Role of HDACs in Regeneration and Ageing in the Peripheral Nervous System: Repair Schwann Cells as Pivotal Targets

**DOI:** 10.3390/ijms23062996

**Published:** 2022-03-10

**Authors:** Jose A. Gomez-Sanchez, Nikiben Patel, Fernanda Martirena, Shaline V. Fazal, Clara Mutschler, Hugo Cabedo

**Affiliations:** 1Instituto de Neurociencias de Alicante, Universidad Miguel Hernández—Consejo Superior de Investigaciones Científicas, 03550 San Juan de Alicante, Spain; npatel@umh.es (N.P.); hugo.cabedo@umh.es (H.C.); 2Instituto de Investigación Sanitaria y Biomédica de Alicante (ISABIAL), 03010 Alicante, Spain; 3Department of Hematology, General University Hospital of Elda, 03600 Elda, Spain; martirena_mar@gva.es; 4John Van Geest Centre for Brain Repair, Department of Clinical Neurosciences, University of Cambridge, Cambridge CB2 0PY, UK; sf618@cam.ac.uk (S.V.F.); cm2019@cam.ac.uk (C.M.); 5Wellcome—MRC Cambridge Stem Cell Institute, Puddicombe Way, Cambridge CB2 0AW, UK

**Keywords:** Schwann cell, repair Schwann cell, HDACs, nerve injury, nerve regeneration, myelin, remyelination, ageing, HDACs therapies

## Abstract

The peripheral nervous system (PNS) has a remarkable regenerative capacity in comparison to the central nervous system (CNS), a phenomenon that is impaired during ageing. The ability of PNS axons to regenerate after injury is due to Schwann cells (SC) being reprogrammed into a repair phenotype called Repair Schwann cells. These repair SCs are crucial for supporting axonal growth after injury, myelin degradation in a process known as myelinophagy, neurotropic factor secretion, and axonal growth guidance through the formation of Büngner bands. After regeneration, repair SCs can remyelinate newly regenerated axons and support nonmyelinated axons. Increasing evidence points to an epigenetic component in the regulation of repair SC gene expression changes, which is necessary for SC reprogramming and regeneration. One of these epigenetic regulations is histone acetylation by histone acetyl transferases (HATs) or histone deacetylation by histone deacetylases (HDACs). In this review, we have focused particularly on three HDAC classes (I, II, and IV) that are Zn^2+^-dependent deacetylases. These HDACs are important in repair SC biology and remyelination after PNS injury. Another key aspect explored in this review is HDAC genetic compensation in SCs and novel HDAC inhibitors that are being studied to improve nerve regeneration.

## 1. Introduction

Epigenetic markers such as acetylation or methylation can change gene expression. Acetylation depends on the opposite action of Histone acetyl transferases (HATs) and Histone deacetylases (HDACs), which acetylate and deacetylate histones, respectively.

HDACs have been shown to play a critical role in peripheral nerve myelination and remyelination following injury. Injured nerves in the peripheral nervous system (PNS) have the capacity to regenerate and reinnervate their target organs, which does not occur in the central nervous system (CNS), due to a lower expression of growth-promoting factors and inhibition by glial cells (reactive astrocytes and oligodendrocytes) [1]. After an injury in the PNS, Schwann cells (SCs) can differentiate into a repair phenotype. This change in the phenotype is controlled by JUN, a transcription factor that is upregulated after injury. In the absence of JUN, SCs are not able to differentiate into this repair phenotype, a phenomenon which also occurs in ageing or chronic denervation [2,3].

HDACs have been implicated in SC development, myelination, and remyelination after injury. Specifically, HDACs are needed to repress the *Jun* promoter and, therefore, allow myelination to proceed [4]. HDAC downregulation is also important in repair SCs, as it has been shown that when certain HDACs are depleted in SCs, remyelination, as well as the induction of the repair program, are affected [5,6].

Furthermore, there is redundancy in HDAC expression, suggesting that there is genetic compensation of HDACs in the PNS: when one HDAC is depleted, another is upregulated, and consequently peripheral nerve myelination and maintenance is conserved [6].

HDACs have also been implicated in several PNS and CNS diseases, such as Charcot Marie-Tooth (CMT), Amyotrophic lateral sclerosis (ALS), or Alzheimer’s disease (AD). HDAC therapies are being studied to improve nerve regeneration, and several studies suggest that HDAC inhibitors (HDACi) can improve disease outcome.

This review focuses on the role of HDACs in SC biology in both development and repair, as well as on genetic compensation between HDACs, a genetic mechanism that is gaining relevance as suggested by recent studies. Finally, we will discuss ongoing clinical trials involving HDACis in the PNS.

## 2. Histone Deacetylases (HDACs)

Chromatin in eukaryotic cells is organized in nucleosomes. DNA is wrapped around histone proteins to form compact nucleosomes, which are defined by an octamer core of histone protein wrapped by 147 base pairs (bp) of DNA. The state of chromatin and, therefore, the accessibility of the genetic code depends on DNA and histone modifications. Methyl residues can be added to DNA, and acetyl groups can be added to histones. The complex formed by DNA, histone proteins, and RNA that packages the genetic information of the cell is called the epigenome [7,8].

Histone acetylation is regulated by two groups of enzymes with opposing functions: Histones Acetyltransferases (HATs), which add an acetyl group, and Histone Deacetylases (HDACs), which remove these acetyl groups. HATs and HDACs control acetylation levels in eukaryotic cells [9]. HDACs most frequently repress gene transcription by forming a complex with transcription factors and other proteins to regulate fundamental cellular processes such as cell cycle progression, survival, and differentiation [10].

Until now, 18 human HDACs have been described, all of which have a highly conserved deacetylase domain [10]. They are divided into four classes based on their homology with yeast deacetylases: class I, class II, class III, and class IV (Figure 1). Class I is composed of HDAC1, HDAC2, HDAC3, and HDAC8. Class II is divided into two groups: class IIa, which includes HDAC4, HDAC5, HDAC7, and HDAC9, and class IIb, composed of HDAC6 and HDAC10. Class III includes Sirtuins (SIRT) 1–7. Class IV is formed only by HDAC11. The enzymatic activity of class I, class II, and class IV HDACs depends on Zn^2+^ ions. In contrast, Class III enzymatic activity depends on nicotinamide adenine dinucleotide (NAD)^+^ levels [11,12]. For this review, we will further describe only HDAC classes I, II, and IV, and discuss the growing evidence for their roles in SC biology and nerve regeneration.

### 2.1. Class I HDACs

HDAC1, HDAC2, HDAC3, and HDAC8 belong to class I HDACs. Class I HDAC expression is ubiquitous, and regulates fundamental cellular mechanisms such as cell proliferation, survival, and migration [10].

Class I HDACs share a structural similarity to yeast RPD3. They have a deacetylase domain with short amino- and carboxy-extensions and are predominantly confined to the nucleus. HDAC1, HDAC2, and HDAC3 form multiprotein complexes, more specifically co-repressor complexes that are regulated by inositol phosphate. Often, HDAC1 and HDAC2 are recruited together, forming the same co-repressor complex, such as the nucleosome remodeling and deacetylase complex (NuRD), Sin3 complex, CoREST complex, or the mitotic deacetylase complex (MiDAC). HDAC3’s enzymatic activity is switched on when it interacts with the deacetylase activation domain (DAD) of NCOR1 and SMRT (also called NCOR2), forming the NCOR/SMRT complex [7,16,18].

### 2.2. Class II HDACs

Class II HDACs have structural homology with yeast HDA1 and have a conserved, inactive deacetylase domain in their C-terminus. They are classified into two subclasses: class IIa, which consists of HDAC4, HDAC5, HDAC7, and HDAC9, and class IIb, which consists of HDAC6 and HDAC10 [19,20].

Class IIa HDACs have a nuclear localization sequence (NLS) which allows them to shuttle to the nucleus, where they exhibit poor enzymatic activity due to the substitution of a catalytically important tyrosine for a histidine [21]. They are only considered to be active when forming a complex with NCOR/SMRT and HDAC3 (which provides the catalytic activity) [18]. All Class IIa HDACs also have conserved serine residues in their N-terminal end, which are important for signal dependent phosphorylation and, therefore, for their localization and function [16,22]. Class IIa HDACs are expressed in the heart, skeletal muscle, brain, and pancreas, and play a significant role in chondrocyte and osteocyte development. They are also relevant for thymocyte differentiation and myocardium maintenance [10].

Class IIb HDACs are mainly cytoplasmic and appear to be important for cell motility. HDAC6 has a microtubule binding domain, and some of its major substrates are non-histone proteins such as α-TUBULIN, Contractin, Ku70, and HSP90 [23]. It also has highly conserved domains that are necessary for cytoplasmic localization, as well as for nuclear export. Nuclear export signaling (NES) mediates translocation to the cytoplasm and interaction with cytoskeletal proteins. HDAC6 is the only class II HDAC with two independent catalytic domains. HDAC10 has a leucine rich domain, and like class IIa HDACs, HDAC10 associates with class I HDACs, forming a multiprotein repression complex [16].

HDAC6 and HDAC10 are expressed in the brain, heart, liver, kidney, pancreas, and spleen [10,24].

### 2.3. Class III HDACs

Class III HDACs are NAD^+^-dependent and are also known as SIRTs, with seven being currently described (SIRT1–7). They have been found in the nucleus, cytoplasm, and mitochondria, and exhibit deacetylase and ADP ribosyl transferase activities that regulate other proteins [18,25].

### 2.4. Class IV HDACs

HDAC11 is the only class IV HDAC. It is equivalent to yeast Hos3 and has homology in its catalytic domain with class I and class II HDACs. Its fatty acid acylase activity is more important than its deacetylase activity [26]. It regulates interleukin 10 expression and the DNA replication factor CDT1 [16]. HDAC11 has a ubiquitous subcellular location, but is predominantly confined to the nucleus [16,27]. It is highly expressed in the brain, but is also present in skeletal muscle, heart, kidney, and testis [27].

## 3. The Role of HDACs in Schwann Cells

Schwann cells (SCs) are the principal glial cells in the Peripheral Nervous System (PNS). They play essential roles in the development, maintenance, function, and regeneration of peripheral nerves.

SCs are derived from neural crest cells, migrate along axons to reach their target tissue, and later differentiate into SC precursors and then into immature SCs between embryonic days 12 and 15 in mice. Around birth, axonal sorting and myelination begin in peripheral nerves. In the mature nervous system, SCs can be classified into two major classes: myelinating and non-myelinating SCs [3,28]. There are, however, also terminal SCs in the neuromuscular junction [29] and nociceptive SCs in the skin [30]. Myelinating SCs provide the myelin ensheathment of all large-diameter peripheral axons or A fibers (relay motor or proprioceptive information, touch, or pressure). Non-myelinating SCs, or Remak SCs, ensheath several small caliber axons or C fibers (respond to pain, thermal, or mechanic stimuli). After axonal sorting, myelinating SCs establish a 1:1 relationship with large diameter axons and wrap around them multiple times to form a thick and compact myelin sheath. This allows the fast conduction of action potentials by insulating the axons [3]. SCs’ decision whether to myelinate or not is dictated by the axon itself, based on the amount of neuregulin 1 type III exposed on its membrane, activating ERBB2 and ERBB3 receptors on SC membranes [31]. The non-myelinating SCs typically associate with several small-diameter axons to form Remak bundles that provide support to the axon. Interestingly, SC differentiation into myelin or non-myelin SCs is reversible during pathology, accounting for the remarkable plasticity of SCs and contributing to the regenerative potential of the PNS [32].

The PNS has a higher regenerative capacity than the CNS. This ability to quickly recover following damage is, to a large extent, due to the plasticity of SCs. Nerve injury triggers the conversion of myelin and non-myelin SCs to a cell phenotype specialized to promote repair, known as repair SCs. These cells provide the necessary signals and spatial cues for the survival of injured neurons, axonal regeneration, and target reinnervation. The conversion into repair SCs involves de-differentiation together with alternative differentiation or activation, a combination that is typical of cell type conversions often referred to as (direct or lineage) reprogramming [33,34]. Thus, injury-induced SC reprogramming involves down-regulation of myelin genes, combined with activation of a set of repair-supportive features, including up-regulation of trophic factors and an increase in cytokines as part of the innate immune response, as well as myelin clearance by activation of myelin autophagy in repair SCs (a process called myelinophagy) [35], and phagocytosis of myelin debris by macrophages and SCs [36,37,38]. Moreover, repair SCs promote macrophage recruitment by LIF (leukemia inhibitory factor) and MCP-1 (macrophage chemoattractant protein-1) secretion, and the formation of regeneration tracks, Büngner bands, that serve as guidance for axons to their targets [32,33,34,35,36,37,38,39]. This repair program is controlled by a transcriptional program involving the transcription factor JUN, which is strongly and rapidly upregulated in SCs after injury. *Jun* elevation in Schwann cells after nerve injury was first reported by De Felipe and Hunt in 1994 [40]. In the absence of *Jun*, damage results in the formation of a dysfunctional repair cell, leading to neuronal death and failure of functional recovery. *Jun*, although not required for SC development, is critical for the reprogramming of myelin and non-myelin (Remak) SCs into repair cells following injury. SC reprogramming is impaired in some diseases or in ageing due to a slow/weak upregulation of some crucial transcription factors, such as *Jun* or *Stat3* [2,3,39,41,42].

Due to their significant role in nerve regeneration, repair SCs and their activation, transcriptome and proteome have been of increasing interest, and have recently also been studied in humans [43,44].

### 3.1. The Role of HDACs during Schwann Cell Development

Several epigenetic modifications involved in the regulation of neural crest development have been described. Regarding HDACs in the SC lineage, it has been shown that class I and class II HDACs have an important role in SC development. In mouse models, the use of conditional knockouts (*Dhh*-Cre, Table 1) has shown that *Hdac1* and *Hdac2* depletion results in massive SC loss, probably through a significant reduction in Sox10 expression [45]. *Hdac1/2* deletion in neural crest cells, using the *Wnt1*-cre mouse, shows that HDAC1/2 activity is necessary for neural crest cell differentiation into SC precursors and satellite glial cells [45]. In this study, authors show that HDAC1/2 bind and induce the *Pax3* promoter. This *Pax3* induction is required to maintain SOX10 levels in developing SCs and satellite glia cells. Also, HDAC1, HDAC2, SOX10, and PAX3 are recruited to the *Mpz* promoter to induce its activation and the expression of MPZ [45]. In addition, SOX10 independently activates the fabp7 promoter and, therefore, regulates FAPB7 protein expression, which is another early marker of the SC lineage [28,45].

### 3.2. The Role of HDACs in Myelination

SCs wrap around axons several times and insulate them to allow saltatory conduction of action potentials. The process of myelination begins after birth in peripheral nerves, when large diameter axons are already sorted in a 1:1 ratio with myelinating SCs and start to be ensheathed by individual SCs spaced along their length. To produce myelin, SCs upregulate myelin genes, and genes encoding the enzymes that synthesize lipids to form the myelin sheath, as well as several proteins that compact and stabilize myelin (*Mpz, Mbp*, *Cnp* and others). This genetic program is determined by several transcription factors, such as SOX10, OCT6, and KROX20, and involves downregulation of negative regulators of myelination, such as JUN, NOTCH, SOX2, or ID2 [34]. It has also been demonstrated that a rise in cAMP induced by the G protein-coupled receptor 126 (Gpr126) controls SC differentiation and the myelination process [50]. Gomis–Coloma et al. (2018) show that Class II HDACs have an important role in SC differentiation and myelination. Cytoplasmic HDAC4 responds to cAMP signaling by shuttling into the nucleus. Once in the nucleus, HDAC4 represses the *Jun* promoter by binding it in a complex with NCOR1 and HDAC3 [4]. In this study, it was also shown that HDAC4 and HDAC5 have a compensatory contribution in the activation of the myelin transcriptional program and myelination process in vivo. Meanwhile, *Hdac4/5* double knockout mice showed elevated *Jun* expression and delayed axonal myelination, consistent with findings using cultured SCs [4]. Moreover, HDAC5 regulates the repression of *Maf* during myelination. *Maf* has been implicated in myelination, as its ablation in myelinating Schwann cells causes hypomyelination. Specifically, HDAC5 is phosphorylated by nuclear CaM-kinases in response to NRG1 signaling and is excluded from the nucleus, increasing *Maf* activation and induction of cholesterol biosynthetic genes [51].

HDAC1/2 also have critical functions during the myelination process. *Hdacs1/2* double cKO mutants are characterized by a radial sorting delay, the absence of myelin, and large amounts of SC apoptosis [15]. In this study, the authors identify specific primary functions for HDAC1 and HDAC2: while HDAC1 maintains SC survival in early postnatal SCs by preventing a precocious increase of active beta-catenin levels, HDAC2 acts together with SOX10 to activate the transcription of *Sox10*, *Krox20*, and *Mpz* and, thereby, induces the myelination program [15]. In a similar study, Chen et al. (2011) show that the absence of HDAC1/2 prevents developmental myelination and leads to low Sox10 expression in SCs, consistent with the study published by Jacob, et al. (2011) [15,52].

HDAC1 and HDAC2 have also been implicated in myelin maintenance in adult nerves. In Hdac1/2 inducible conditional knockouts using inducible *Mpz*-Cre^ERT2^, motor and sensory functions are impaired, and there is a decrease in *Mpz* expression. These decreased levels of MPZ cause impaired nodal and paranodal structures with lower levels of NFacs186, NFasc155, and Caspr [46]. In addition, authors show that in this inducible conditional knockout (with *Plp1*-Cre^ERT^), demyelination in vitro in SC co-cultures with dorsal root ganglion (DRG) neurons, is decreased when exogenous MPZ is expressed in SC [46].

Two studies have identified HDAC3 as an inhibitor of developmental myelination, allowing the shift to a homeostatic myelination program which is responsible for myelin maintenance in adults nerves. The loss of this stability is associated with neuropathology [5,49]. HDAC3 inhibition during development promotes *Krox20* expression, resulting in hypermyelination. Additionally, HDAC3 antagonizes Neuregulin–PI3K–AKT signaling in myelination. In *Hdac3* conditional knockouts, under *Cnp*-cre (*Hdac3* cKO) or tamoxifen inducible *Plp1*-CreERT (*Hdac3* iKO), there is an increase in promyelinating transcription factors and myelin gene expression, causing hypermyelinated axons during development and in mature nerves [5]. In *Hdac3* conditional mutant mice, under *Mpz*-cre, it has been shown that HDAC3 is a regulator of myelin gene expression, and HDAC3 loss of function in SC results in hypermyelination in adult nerves [49]. Further studies are, however, needed to elucidate the role of HDAC3 in development and myelin maintenance in SCs.

In addition, other studies show that HDAC1, HDAC2, and HDAC3 form a complex together with Schwann cell factor 1/positive regulatory domain protein 4 (SC1/PRDM4) [53]. This complex represses *Cyclin E* transcription through the binding of SC1/PRDM4 to the *Cyclin E* promoter and causes cell proliferation arrest. SC1 is a p75NTR-interacting zinc finger protein [53]. The role of this complex has, however, not yet been studied in vivo.

HDAC1 and HDAC2 have also been implicated in myelination as part of the NuRD complex. Along with SWI/SNF-like ATPase subunit (CHD3 or CHD4, chromodomain helicase DNA-binding protein), they can reposition nucleosomes. A SC-specific deletion of *Chd4* caused a transient delay in myelination, and demyelination in mature nerves [54]. The *Nab1/2* and *Chd4* knockouts, however, did not show as profound of a loss of SOX10 and KROX20 transcription factors as the *Hdac1/2* double knockout did. While the NuRD complex was originally thought to be primarily repressive, either CHD4 or the associated NuRD complex can also activate transcription. Accordingly, the CHD4 conditional knockout nerves showed deficient repression of Krox20 target genes that are normally repressed during SC development, but also reduced activation of some major myelin genes [54].

Histones are not the only target of HDACs, as they also regulate transcription factors like NF-κB. HDAC1 and HDAC2 modulate the acetylation state of NF-κB, a process that is critical to switch on the myelination program in SCs. HDAC1/2 mutants, using *Dhh*-cre mice, and *Hdac1* and *Hdac2* floxed mice, present heavily acetylated NF-κB-p65, blocking the expression of positive regulators of myelination such as *Sox10*, *Oct6*, and *Krox-20 (Egr2)*, and inducing the expression of differentiation inhibitors, such as *Sox2*, *Sox11*, *Jagged1*, *Jun* (*c-Jun*), *Hes1*, *Hes5*, *Id2*, and *Id4* [52]. HDAC1 also directly interacts with SOX10, and this appears to be required for gene activation and maintenance of *Mpz* expression even in mature SCs [15,46].

While there are no in vivo studies in the PNS involving HDAC11, the unique class IV HDAC, there are some developmental studies in oligodendrocytes, the myelinating glial cells in the CNS. Specifically, Hdac11 KO (null KO) mice show reduced *Mbp* and *Plp* expression due to a reduction of acetylation levels in histone H3 at residues lysine 9 and lysine 14 (H3K9/K14ac) [55].

### 3.3. Hdac Expression during Development and Myelination

Recent studies using RNA sequencing (RNA-seq) and single cell RNA-seq granted the opportunity to review gene expression in peripheral nerves at different time points in different cell types. The Sciatic Nerve Atlas (SNAT) webpage from the University of Zurich allows simple and quick checks of gene expression in the SC lineage [56]. The description of gene expression in SNAT during nerve development (from E13.5 to P60), has become an important tool for the identification of important genes and the discrimination of cell type specific expression (Figure 2).

All HDAC subtypes are expressed in nerves. Class I *Hdacs*, which include *Hdac1*, *Hdac2* and *Hdac3*, are mostly expressed during development and at P60. *Hdac6* and *Hdac11* are predominantly expressed during embryonic development (E13.5) after which expression decreases significantly. Similarly, *Hdac4* is mainly expressed during embryonic development, although expression is significantly upregulated between E13.5 and E17.5 and then again significantly downregulated by P1. Gerber et al. further describe in their single cell sequencing data (see Figure 3) that SCs are not the main cell source for *Hdac4*, *Hdac6, Hdac7*, and *Hdac11* gene expression levels during development, with fibroblast-related cells and endothelial cells contributing. Expression levels of these *Hdacs* are, overall, not as high as *Hdac1*, *Hdac2* and *Hdac3*, which are highly expressed during embryonic development and then reach significantly lower levels by P14, that are maintained into adulthood. *Hdac5* gene expression is significantly upregulated during early embryonic development (E13.5), and then drops before birth (E17.5). After birth, *Hdac5* and *Hdac7* are both gradually, but significantly, upregulated until P60. SCs are, however, also not the main cell source contributing to the expression of these HDAC subtypes, with especially pericytes/endothelial cells contributing (Figure 3) [56].

*Hdac8* and *Hdac10* show low expression levels overall, while *Hdac9* is undetectable in nerves after birth. These *Hdac* subtypes are mainly expressed in epineurial, perineurial, and endoneurial cells (Figure 3) [56].

### 3.4. The Role of HDACs in Repair Schwann Cells and Remyelination

To understand the role of HDACs in nerve regeneration and repair SC reprograming, the use of HDAC inhibitors and genetically modified organisms, predominantly mutant mice, is essential. Kim et al. (2019) used a reversible HDAC class I and II inhibitor, Trichostatin A (TSA), to study the role of HDACs in nerve regeneration. TSA is an antimicrobial drug extracted from *Streptomyces hygroscopicus*, which acts as a noncompetitive enzyme inhibitor. It effectively suppresses SC reprograming after nerve injury, and inhibits myelin and axon clearance, as well as the reprogramming and proliferation of SCs [57]. The application of TSA to regulate HDAC levels in SCs could consequently be an effective way to delay peripheral neurodegeneration (Table 2).

The class I HDAC inhibitor valproic acid (VPA), which is extensively used as an antiepileptic drug, has been shown to improve nerve regeneration as well. It selectively induces proteasomal degradation of HDAC2. In rats, when VPA conduits (silicon tubes coated with VPA) are used following sciatic nerve injury, regeneration and functional recovery are improved [58,59]. A recent study using bioabsorbable conduits (hydroxyapatite/poly d-l-lactic acid, PDLLA) with sustained release of VPA in injured rats has shown improved peripheral nerve regeneration that is comparable to autografts. These autografts are regularly used to repair peripheral nerve injuries and have better results than empty conduits, but they have the disadvantages of limited availability and donor site morbidity [60]. These studies using TSA and VPA are of particular interest, as they show that inhibition, degradation, or knockout of class I HDACs does not necessarily have the same effect on regeneration. This can be explained by contextual differences in animal models, different mechanisms of action of inhibitors used, subcellular localization and post-translational modifications of HDACs, or different substrates or diverse binding proteins [61].

In vivo studies, using conditional knockout mice of HDACs class I and II, support HDACs playing a major role in repair SC regulation and nerve regeneration. Brügger et al. (2017) describe how HDAC1 and HDAC2 are strongly upregulated in SC after nerve injury and show that, in *Hdac1/2* dKO (using *Mpz*-Cre^ERT2^), remyelination is impaired after injury in the adult [47]. They identified that SUMOylated HDAC2 is responsible for the early injury response in SCs. Specifically, Sox10, in coordination with HDAC2, recruits other chromatin-remodeling enzymes (demethylases), such as JMJD2C and KDM3A, to activate Oct6 and Krox20 genes (sox10 targets) after nerve injury [47].

Another study with class I HDACs shows that a class I HDAC knockout mediated reduction in Oct6 levels results in faster conversion into repair SCs, increased myelin clearance, and improved axonal regrowth after injury [62]. To stimulate early remyelination in wildtype C57BL/6 mice, Brügger et al. (2017) used a short treatment with Mocetinostat, a HDAC1/2 inhibitor, which increased regeneration and remyelination after injury [47]. They also documented that by stimulating eEF1A1 deacetylation, myelination is stimulated. Acetylated eEF1A1 translocates to the nucleus where it binds to Sox10 and transports it out of the nucleus. By stimulating eEF1A1 deacetylation, Sox10 remains inside the nucleus where it targets promyelinating genes increasing the remyelination process in PNS and CNS [48].

Conditional or inducible knockout of *Hdac3* enhances SC remyelination and myelin thickness after injury [5]. These observations suggest that HDAC3 inhibition accelerates SC differentiation, enhances remyelination, and improves functional recovery in repair SCs. Inhibitors such as sodium phenylbutyrate (PBA) decrease inflammation induced by SC through the inhibition of HDAC3 activity and regulation of protein expression via the NFκB-p65 pathway [63]. When PBA is used in injured mice, axonal regeneration is stimulated, as well as remyelination. When HDAC3 protein levels decrease, nuclear translocation of NFκB-p65 is induced, reducing the levels of pro-inflammatory cytokine expression (such as TNFα, IL-1β, and IL-6) in SCs. These findings support the idea that HDACs play a role in the transcriptional regulation of pro-inflammatory cytokines [63] and suggest a potential application for HDAC3 inhibition in improving peripheral nerve regeneration.

Recent work carried out by Velasco–Aviles et al. (2022) focusing on class II HDACs has shown that they also play an important role in repair SC biology and remyelination, in particular *Hdac4*, *Hdac5*, and *Hdac7*, which are expressed in SCs [6]. In a SC-specific triple cKO of these HDACs, using *Mpz*-Cre mice, they saw that remyelination is impaired, but repair SC activation is accelerated due to increased myelin clearance in mutant mice after injury. The same effect can be observed in development in the *Hdac4/5* dKO or the *Hdac4/5*/7 triple knockout (tKO). *Hdac4/5/7* tKO mice show elevated JUN levels after nerve injury, and this JUN upregulation has been shown to delay remyelination in nerve regeneration, the same delay seen in *Jun*_OE overexpressing mice [64]. In injured nerves of single cKO for *Hdac5* and *Hdac7*, Velasco et al. (2022), did not find a significant effect on remyelination or in repair SC activation, and in *Hdac4* single cKO there was only a slight delay in remyelination, but not in repair SC activation [6].

To date, no nerve regeneration studies have been performed with Hdac11 knockouts (class IV HDAC) or a specific class IV drug such as elevenostat (JB3-22) [26,65].

Further to the role of HDACs in SCs, levels of histone acetylation in axons have vital roles after injury and in repair (reviewed in [66,67,68,69]). In sensory neurons, injury produces a calcium wave that induces the PKCγ-dependent nuclear export of HDAC5, and consequently axon regeneration is accelerated [70]. HDAC5 is also transported to the growth cone in injured axons, which modulates growth-cone dynamics to sustain axon regeneration [70]. In addition, it has been described that calcium-dependent activation of PP4/2 signaling controls the axonal regenerative ability via HDAC3 inhibition [71]. Furthermore, HDAC6 inhibition increases growth cone size in sensory axons and increases axonal acetylated Miro1, which prevents chondroitin sulfate proteoglycans-dependent decreases in mitochondrial transport and sensory axon growth [72].

A further factor determining the effect of HDACs during myelination and remyelinations is the dependence of myelin on high amounts of fatty acids for its assembly and maintenance. SCs are consequently especially vulnerable to lipid synthesis dysregulation. Acetyl-CoA (acetyl coenzyme A) is a crucial metabolite for this, as it is at the crossroads of lipid metabolism and energy generation. It is the substrate of HATs and HDACs, and, therefore, inhibition of HDAC activity, both genetically or pharmacologically, could impact on acetyl-CoA metabolism. Specifically, it could create a metabolic deficit of Acetyl-CoA in SCs that could affect fatty acid synthesis, impacting myelination. While regulation of SC fatty acid synthesis has been described in several papers, with studies focusing on the synthesis of the fatty acids precursor, acetyl-CoA, there are no studies which use *Hdac* mutant mice or pharmacological inhibitors that consider Acetyl-CoA levels [73,74,75,76,77]. Myelination or remyelination impairment in some HDAC knockouts could, however, be caused by Acetyl-CoA dysregulation in the SC and it would, consequently be interesting to explore this in future studies using *Hdac* or *Hat* mutant mice.

### 3.5. HDACs in Ageing or Disease

The PNS, too often the ignored younger cousin of the CNS, is a wonderfully intricate system composed of neurons, glia, immune, and connective cells, all operating in concert to provide some functions in common with the brain, and some, such as the capacity to regenerate, that are unique. Given this complexity, it is not surprising that many age-associated disorders of the PNS are well described and yet sorely lacking in mechanistic insight. For example, the incidence of peripheral neuropathies increases rapidly with advancing age, with some estimates suggesting that nearly one third of the population over the age of 65 suffers from some form of neuropathy [78]. The consequences of peripheral neuropathies are disabling, commonly manifesting as sensations of burning, tingling, or numbness in the extremities. Furthermore, the accompanying loss of balance can contribute to traumatic falls in the elderly [79,80,81].

In the aged injured PNS, the interaction between SC and regenerative axons takes longer, and myelin clearance, macrophage recruitment, and the amount of trophic factors secreted by repair SC and target organs decreases. Recently, five separate studies focussing on the ageing PNS in murine systems came to a similar, surprising conclusion: aged SC or macrophage dysfunction, and not neuronal dysfunction, ultimately restrains the re-growth of axons after nerve injury [41,82,83,84,85,86,87]. In addition, it has been demonstrated that age-associated or long-term denervated stumps in injured mice show defective SCs with impaired nerve regeneration after injury [87,88,89]. This deterioration of nerves distal to the injury site has been modelled and documented in rodents and is considered a major reason for regeneration failure in humans [84,85,87,90]. The mechanisms that control the phenotypic instability of repair cells are, therefore, attractive therapeutic targets, as well as of great biological interest, yet so far unexplored. Aged SC have a reduced capacity to be reprogramed into repair SC. Downregulation of myelin genes is impaired, and repair associated gene upregulation is also decreased. There is a poor upregulation of *Jun* in aged SC after injury [41]. When this diminished *Jun* upregulation is restored in SC, regeneration in aged nerves is improved [87]. Recently, it has been described that the transcription factors JUN and STAT3 play a fundamental role in the long-term maintenance of repair or ageing SCs [87,88]. In summary, aged repair SCs have a reduced ability to induce *Jun* activation, a decreased exogenous myelin debris ingestion, and a low cytokine expression after nerve injury [42,84]. It has also been shown that repair SC length diminishes in long-term denervated nerve stumps [91].

During the last 20 years, it has become increasingly clear that epigenetics, including DNA methylation, histone modifications, non-coding RNAs, and microRNA, plays a crucial role in the ageing process [92]. Histone modifications, including methylation and acetylation, have been intimately linked to lifespan regulation [93]. These modifications are reversible, and their reversal could consequently be used as a tool to improve ageing dependent degeneration and increase lifespan. The possibility for lifespan regulation is best characterized in this regard, mainly due to the advent of HDAC inhibitors from the cancer biology field [94]. In yeast, *C. elegans* and *Drosophila Melanogaster* it has been shown that class II HDAC inhibition or deletion extends lifespan through trehalose metabolism (a conserved storage carbohydrate) [95]. The function of Class II HDACs in ageing is relatively unknown and the existing data is sometimes contradictory. For example, in a Huntington’s disease model, the reduction of HDAC4 is shown to facilitate clearance of pathogenic huntingtin protein that aggregates and enhances neuron degeneration [96]. In contrast, another study reported HDAC4 overexpression delaying cellular senescence via SUMOylation of the class III HDAC SIRT1, whereas HDAC4 knockdown leads to premature senescence in human fibroblasts [97]. Class I HDACs such as HDAC1 may also play an important role in ageing. In the ageing brain, in a model of Alzheimer’s disease, HDAC1 deficiency causes impaired OGG1-initated 8-oxoguanine activity, an enzyme that is responsible for repairing oxidative DNA damage in the brain [98]. Additionally, the role of HDAC11 has been investigated in multiple sclerosis, schizophrenia, learning, and memory [99,100,101].

Although the role of HDACs in regeneration in the aged PNS has been poorly studied, it has been broadly studied in cancer, neurodegeneration, cardiometabolic diseases, liver dysfunction, sarcopenia, inflammation, and arthritis, as well as in diseases characterized by premature ageing in preclinical mouse studies. For some of these ageing-related diseases (such as liver dysfunction or sarcopenia), evidence for HDAC inhibitors as an effective treatment is only now emerging. There is substantial evidence for a role of HDACs in many physiological processes and, therefore, potential therapeutic applications are being investigated in a range of ongoing clinical trials in various fields, including neurodegeneration and regeneration [102].

### 3.6. Hdac Levels after Injury and Ageing

Using RNA-seq data from a nerve injury study [87], it is possible to gain insight into the expression of *Hdac* genes after nerve injury in young and aged mice, and in chronic injury in young mice (long-term nerve injury without axonal regeneration). While all *Hdacs* are expressed to some extent after injury, the expression levels of some of them are low (Figure 4).

In acutely injured young nerves (P60), class I HDACs, *Hdac1*, *Hdac2*, and *Hdac3*, are the most highly expressed after injury. At 3 days post-injury, both *Hdac1* and *Hdac2* are up-regulated significantly, and at 7 days these HDACs are down-regulated to levels similar to the uninjured nerve (Figure 4A). The class II HDACs *Hdac9* and *Hdac10* are also up-regulated significantly, but expression is lower than that of class I HDACs. *Hdac11* is down-regulated significantly after injury.

In chronic lesions, where distal stumps have no axonal contact for a long period of time, some *Hdacs* are persistently upregulated for up to 10 weeks. In particular, *Hdac5* and *Hdac7*, which are not upregulated in acute injury, show slight but not significant upregulation, and *Hdac11* is upregulated significantly. By contrast, *Hdac6* and *Hdac8* are significantly downregulated in chronically injured nerves (Figure 4A). It is remarkable that in chronic injuries, several *Hdacs* are expressed differently 70 days after injury than in acute injuries. This indicates that a precise modulation of *Hdac* expression during acute and chronic injury is important, and it should be considered in chronic neurodegenerative contexts from a therapeutic perspective. It would also be interesting to check *Hdac* levels and the effect of different inhibitors in this context.

In aged nerves (12-month-old mice), the up-regulation of *Hdacs* 3 days after injury is very similar to that in young, injured nerves, and levels are not significantly different between young and old mice 3 days after injury. Specifically, the upregulation of the class I HDACs, *Hdac1* and *Hdac2,* and the downregulation of the class IV HDAC, *Hdac11,* is significant in aged mice as well. Interestingly, while the upregulation of *Hdac9* and *10* is significant after acute injury in young mice, it is not in old mice (Figure 4B). Overall, all *Hdacs* have a similar pattern of regulation in young and aged mice, and similar expression levels in uninjured nerves, with only *Hdac9* levels being significantly elevated in uninjured older mice.

## 4. HDAC Genetic Compensation in Repair Schwann Cells

Genetic compensation, or genetic buffering, is a process by which a cell, tissue, or organism with a pathogenic mutation does not develop the expected adverse phenotype due to gene duplications or compensatory actions of another gene or genes, which functionally compensate for the loss-of-function genotypes, restoring normal function of the mutated gene and maintaining a normal physiology [103]. Almost 90 years ago, the first genetic compensation was reported in *Drosophila Melanogaster*, and further compensations have since then been reported in other organisms, such as plants (*Arabidopsis thaliana*), yeasts (*Saccharomyces* sp.), zebrafish (*Danio rerio*), mice (*Mus musculus*), and humans (*Homo sapiens*) [104].

Genetic compensation can occur at the transcriptional level. Thus, the observation that some mutations can trigger the transcriptional modulation of other genes, a process also named transcriptional adaptation, has provided a novel explanation for the contradictory phenotypes observed in knockdown versus knockout models and has also increased awareness of the use of these technologies [105]. Different studies have predicted that redundancies are evolutionarily unstable and only have a transitory lifetime. Despite this, there are many examples of transcriptional adaptations or functional redundancies that have been conserved during evolution [104,105,106].

Gene dosage fluctuations (noise) are a well-known phenomenon that has been studied from bacteria to mammalian cells and that could have a significant influence on the fitness of an organism [107]. It has been hypothesized that gene redundancy could have been selected to overcome the negative effects of gene expression noise [106]. Therefore, an adverse effect of a possibly reduced expression of a noisy gene which is fundamental for a certain biological process (such as differentiation) can theoretically be mitigated by the expression of a paralogous gene, a redundant gene controlled by a different promoter [105].

Several studies have reported that genetic compensation of HDACs occurs in different tissues. In adult skeletal muscle, for instance, four class IIa *Hdacs* alleles (*Hdac4*, *Hdac5, Hdac7*, and *Hdac9*) need to be deleted to observe an increase in slow-fiber gene expression. These class IIa HDACs normally repress the transcription factor family myocyte enhancing factor 2 (MEF2), regulating slow fiber caliber [108].

In brain development, genetic compensation of HDACs also occurs. The combined deletion of *Hdac1* and *Hdac2* results in severely impaired brain development and embryonic lethality. This can be prevented by a single allele of either *Hdac1* or *Hdac2*, suggesting that both HDACs have the capacity to compensate and rescue normal cellular function required in the brain [109]. Additionally, a similar compensatory effect between *Hdac4* and *Hdac5* has recently been reported in learning and memory [110].

Genetic compensation is also important during peripheral nerve development. It has been shown that *Hdac1* and *Hdac2* function redundantly during SC development [15,52]. Indeed, ablation of *Hdac*1/2 in SCs leads to impaired myelination during development as well as impaired remyelination after lesion. Despite this, no phenotype was found in single *Hdac1* or *Hdac2* mutants [47]. In the double KO mice, *Sox10* levels were normal in the repair SCs, but *Krox20* upregulation was strongly reduced during remyelination. In addition, SUMOylated HDAC2, JMJD2C, and KDM3A collaborate in repair SCs to sequentially de-repress *Oct6* and *Krox20*, two SOX10 target genes that are critical for SC development and repair SC reprogramming after nerve injury [47].

In Class IIa HDACs, genetic compensation has recently been reported as well. Velasco-Aviles et al. (2022) describe genetic redundancy in repair SCs (as well as in development, see Section 2.2) using cKO in *Mpz*-Cre mice [6]. In regeneration and during remyelination, single cKO of *Hdac4*, *Hdac5*, or *Hdac7* does not have any effect on remyelination. When both *Hdac4* and *Hdac5* were depleted in *Hdac4/5* dKO mice, both myelination during development and remyelination after injury were delayed. Interestingly, in these mice, the transcription factor JUN induces the compensatory overexpression of *Hdac7*, which can partially compensate for the absence of *Hdac4* and *Hdac5.* Indeed, when all three genes were removed from SCs (*Hdac4/5/7* tKO) myelin development and remyelination after injury was impaired for a significantly longer period of time. Strikingly, *Hdac9*, the fourth class IIa HDAC, which is not normally expressed in nerves, is highly upregulated in tKO nerves, which could partially compensate for the absence of other class IIa HDACs. *Hdac9* de novo expression is induced by the expression of MEF2D, which binds to the *Hdac9* promoter activating its expression. In addition, in this tKO mouse, the authors show that myelin clearance after injury is accelerated [6].

HDACs genetic redundancy, however, does not always happen or does not necessarily have a compensatory effect to prevent the appearance of a pathologic phenotype. For instance, *Hdac3* depletion in the liver leads to increased hepatocellular damage and disrupts metabolic homeostasis; however, in these mice, *Hdac1* or *Hdac2* do not show compensatory upregulation [111].

Several previous studies investigating class I or class II HDACs do not consider that one HDAC gene depletion could be upregulating another paralogue HDAC gene to compensate for this deletion. Most studies using HDAC inhibitors do not take into account the putative compensatory overexpression of another HDAC when drawing their conclusions [112]. In any future studies where HDAC activity is blocked genetically or pharmacologically, the process of compensation should always be considered.

## 5. HDACs Therapies: Approved, Trials and Future Directions

Over the last three decades, at least 30 HDAC inhibitors (HDACi) have been extensively studied in clinical trials, several of which have already received regulatory approval. They have been shown to be effective in a broad range of diseases, especially in cancer treatment, where they have been mainly applied to treat hematological malignancies. HDACi have also been shown to be potential treatments for metabolic diseases such as diabetes, neurodegenerative diseases such as Multiple sclerosis, viral infections such as HIV, and peripheral neuropathies such as Charcot–Marie–Tooth disease (CMT) [113,114].

Scientific publications and clinical trials around this topic are rapidly increasing. To date, at least 700 clinical trials are registered and more than 200 are currently recruiting participants. From 2011 to 2020 at least 2000 HDACi papers were published per year, with patents representing 6–11% of the total publications [113]. The global market for HDAC research has been predicted to grow by 32% annually [114].

Five of these agents with deacetylase mechanisms of action have already been approved by regulatory agencies: Vorinostat, Belinostat, Romidepsin, Tucidinostat, and Panobinostat. Vorinostat, a hydroxamic acid derivate (SAHA, suberoylanilide hydroxamic acid) was the first HDACi to be approved by the FDA in 2006 for the treatment of cutaneous T cell lymphoma (CTCL). Romidepsin was approved in 2009 for CTCL as well, Belinostat was approved in 2014 for the treatment of peripheral T cell lymphoma (PTCL), and Parabinostat received FDA approval in 2015 for Multiple Myeloma treatment. Tucidinostat, a non-hydroxamic benzamide class HDAC was approved by China’s National Medical Products Administration for PTCL in 2014 and for postmenopausal advanced breast cancer in 2019 [114].

The molecular targets for all approved inhibitors are classical HDACs (class I, II, and IV), as dysfunctional expression of classical HDACs is associated with tumorigenesis and a higher tumor burden. Class I HDACs and class III HDACs have tumor suppressor functions: conditional knockout mice for Hdac1 and Hdac2 develop lymphoid malignancies [115], as well as non-hematopoietic tumors.

The FDA-approved HDACis, originally approved for the treatment of hematopoietic cancers, are currently also being studied for non-hematological malignancies. HDACis are also being investigated in a range of other diseases due to their well-documented roles in human pathology. These include viral silencing, and HDACi have been widely tested in clinical trials for their ability to reverse HIV latency. They have also been investigated as a potential therapeutic avenue to treat diabetes as HDACs contribute to glucose homeostasis in mammals [116,117,118,119,120]. Immune-mediated diseases such as chronic obstructive pulmonary disease, rheumatoid arthritis, and inflammatory bowel disease are further examples of diseases where HDACi treatment is being investigated [121,122].

Importantly, it has been well-documented in animal models that HDACs play a significant role in the neuropathology of the CNS, particularly in conditions such as Multiple Sclerosis, Amyotrophic Lateral Sclerosis (ALS), Parkinson’s disease, Alzheimer’s disease (AD), Huntington’s disease, frontotemporal dementia, in psychiatric conditions such as Schizophrenia or mood disorders such as depression or anxiety [69,123].

In the last decade, growing evidence has also shown that HDACs have a significant impact on the development of PNS disease. HDAC6 has been implicated in peripheral neuropathy, neuropathic pain, in CMT, and in chemotherapy-induced peripheral neuropathy. The proposed mechanism for pathogenesis is mitochondrial dysfunction, altered mitochondrial transport, and microtubule instability [124]. The study of HDACis in diabetic neuropathy is yielding very promising results [112,125]. Ricolinostat, an oral, selective HDAC6i is being investigated in a phase 2 clinical trial for diabetic neuropathic pain (see Table 3) [126].

HDACs also play an important role in the adult nervous system and in degeneration. HDAC2 plays a role in synaptic plasticity, dendritic spine density, and memory formation [127]. HDAC2 dysregulation is also associated with neuronal degeneration during AD progression [128]. HDAC6 inhibitors have been shown to be of potential benefit in neurodegeneration in ALS, as they can restore histone acetylation levels and consequently slow down disease progression in a FUS (fused in sarcoma) mouse model of ALS [129]. HDAC6 is also a potential target for peripheral neurodegenerative neuropathy, as it has been demonstrated that increases in deacetylated α-TUBULIN and dysregulated axonal transport could be a common pathogenic mechanism of various genetic forms of CMT [130].

Two selective HDAC6 inhibitors, ACY-1215 (Ricolinostat), and ACY-241 (Citarinostat) are currently being studied in clinical trials for hematologic and non-hematologic cancers [130]. Interestingly, ACY-1215 has also been shown to restore nerve damage and reduce pain, numbness, and muscle weakness resulting from chemotherapy and CMT [126,131,132]. A Phase II trial of ACY-1215 for the treatment of diabetic neuropathic pain is currently in the recruitment phase.

The currently approved treatments are all pan HDAC inhibitors, and this lack of specificity is responsible for common side effects such as fatigue, cytopenia, nausea, and, in some cases, cardiotoxicity. Future research needs to be more focused on selective inhibition of HDAC subtypes and future efforts should focus on describing new molecules with increased tissue specificity.

## Figures and Tables

**Figure 1 ijms-23-02996-f001:**
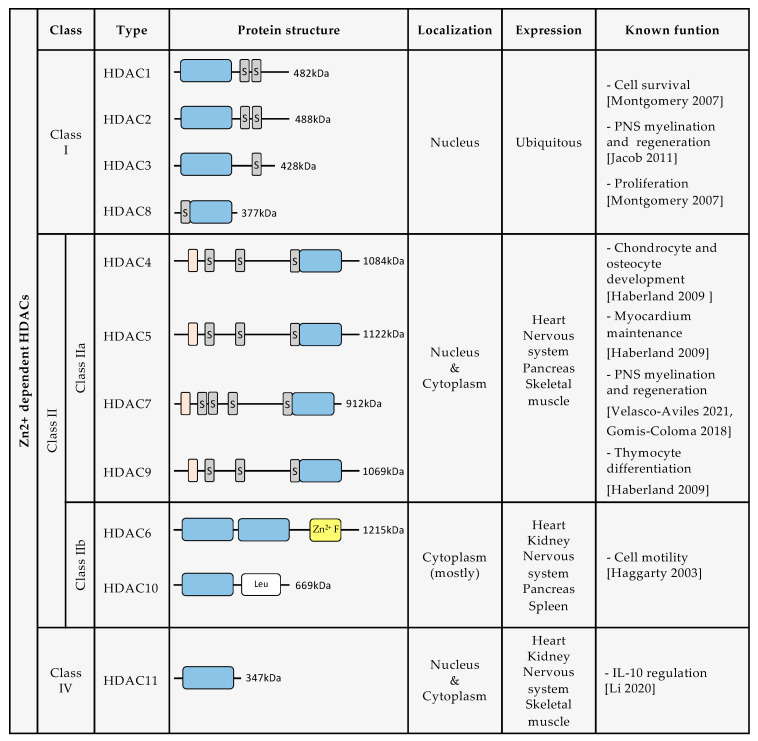
Description of protein structure, localization, expression, and known biological functions of Zn^2+^ dependent HDACs [4,6,13,14,15,16,17]. (Dark grey rectangles containing an S indicate serine residues. Orange rectangles indicate a MEF2 binding site. Blue rectangles indicate the deacetylase domain. Yellow rectangle indicates a Zn^2+^ finger. White rectangle indicates leucine rich domain).

**Figure 2 ijms-23-02996-f002:**
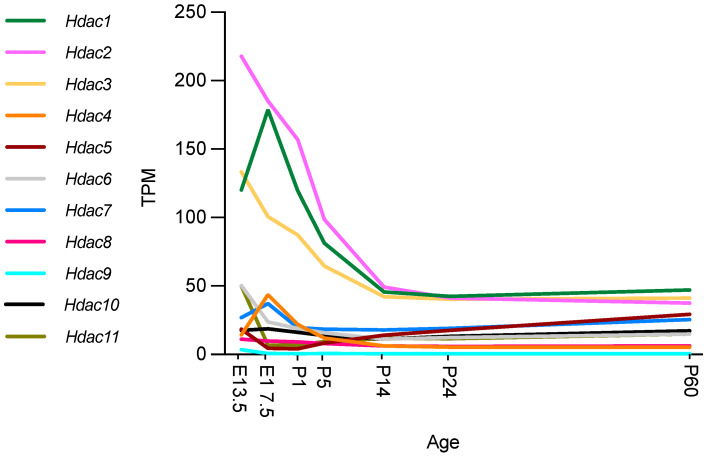
*Hdac* expression during mouse nerve development. Expression in transcripts per million (TPM). Graphs ggenerated based on RNA-seq data from Gerber et al., 2021, GEO database accession number GSE137870 [56]. Authors used four independent samples for bulk RNA sequencing at every time point, two from male and two from female mice. At E13.5 and E17.5, sciatic nerves were pooled from two embryos of the same sex per sample. At all postnatal time points examined (P1, P5, P14, P24, and P60), sciatic nerves from one mouse were used for each independent sample [56].

**Figure 3 ijms-23-02996-f003:**
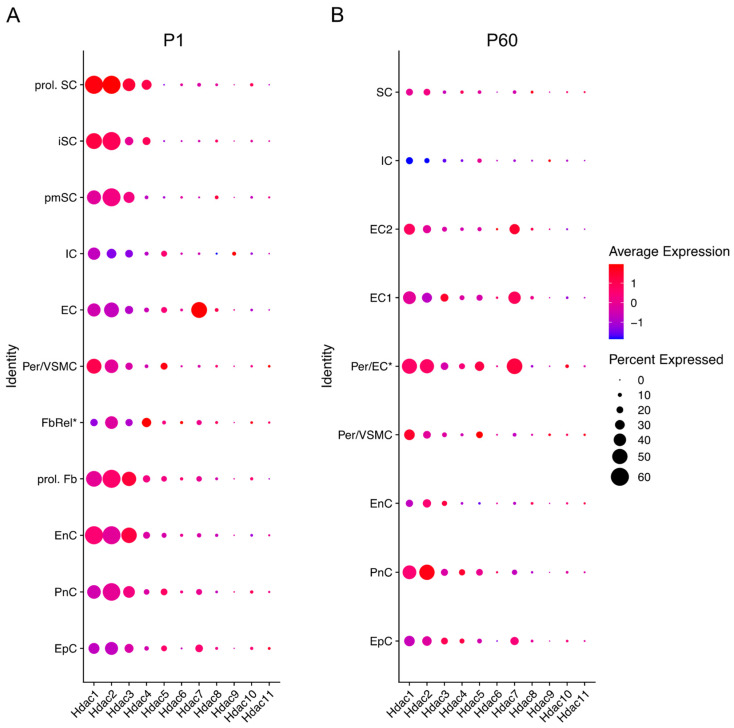
*Hdac* expression across different cells in the sciatic nerve at P1 (early developmental stage) and P60 (mouse adulthood). Data based on 10× genomics single cell RNA sequencing data from Gerber et al. (2021). Graphs generated based on RNA-seq data from Gerber et al., 2021 (GEO database accession number GSE137870). [56]. Three independent samples were included for each time point, processed as three independent 10× Genomics runs. Expression is detailed in proliferating (prol. SC), immature (iSC), and pro-myelinating (pmSC) Schwann cells; immune cells (IC), endothelial cells (EC), pericytes, and vascular smooth muscle cells (per/VMSC), pericytes/endothelial cells (Per/EC*), endothelial cells 1 and 2 (EC1&2), proliferating fibroblast-like cells (prol. Fb), fibroblast-related cells (FbRel*), endoneurial cells (EnC, also known as endoneurial fibroblasts or fibroblast-like cells), perineurial cells (PnC), and epineurial cells (EpC). Dot colour represents average expression level in cells expressing the *Hdac* of interest. Dot size represents percentage of cells expressing the *Hdac* of interest. (* indicates tentative label suggested by Gerber et al., 2021). Futher single cell RNA-seq gene expression can be checked in the Sciatic Nerves Atlas: https://snat.ethz.ch/index.html (Accessed on 26 February 2022).

**Figure 4 ijms-23-02996-f004:**
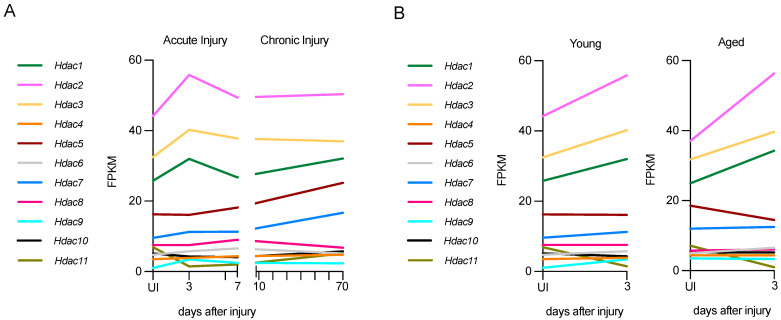
*Hdac* expression in acute and chronic injury in young nerves or acute injury in aged nerves. (**A**) *Hdac* expression in young nerves, both in acute (up to 7 days) and chronic injury (up to 70 days). (**B**) *Hdac* expression in acute injury (3 days) in young and aged nerves. Expression in fragments per kilobase of exon per million mapped fragments (FPKM. Graphs generated based on RNA-seq data from Wagstaff et al., 2021 (ArrayExpress ID E-MTAB-9640) [87].

**Table 1 ijms-23-02996-t001:** Conditional or tamoxifen inducible knockout mice used to study the role of HDACs in Schwann cells (SCs). MGI ID number from the Mouse Genome Informatics (http://www.informatics.jax.org/) (Accessed on 26 February 2022).

*Hdac* Mutant Mouse	Cre Recombinase (MGI ID Number)	Genetic Mutation (MGI ID Number)	Effects on Schwann Cell Biology	Paper
*Hdac1* cKO	*Dhh*-cre (MGI:4359600)	*Hdac1*^flox/flox^ (MGI:4440556)	SC loss	[15]
*Hdac2* cKO	*Dhh*-cre	*Hdac2*^flox/flox^ (MGI:4440560)	Decreased Sox10 and Krox20 levels	[15]
*Hdac1/2* dKO	*Dhh*-cre	*Hdac1*^flox/flox^ (MGI:4440556) *Hdac2*^flox/flox^ (MGI:4440560)	Radial sorting delay Impaired Myelination Reduced *Sox10* and *Krox20* levels Increased *Oct6* levels	[15]
*Hdac1/2* dKO	*Wnt1*-cre (MGI:2386570)	*Hdac1*^flox/flox^ (MGI:4440556) *Hdac2*^flox/flox^ (MGI:4440559)	Impaired SCP differentiation from neural crest Low *Sox10* levels	[45]
*Hdac1/2* idKO	*Plp*-cre^ERT2^ tamoxifen-inducible (MGI:2663093)	*Hdac1*^flox/flox^ (MGI:4440556) *Hdac2*^flox/flox^ (MGI:4440560)	Demyelination in in vitro cocultures with DRG neurons	[46]
*Hdac1/2* idKO	*Mpz*-cre^ERT2^ tamoxifen-inducible (MGI:2663097)	*Hdac1*^flox/flox^ (MGI:4440556) *Hdac2*^flox/flox^ (MGI:4440560)	Impaired Motor and sensory functions Demyelination and decreased *Mpz* expression Impaired nodal and paranodal structures	[46]
*Hdac1/2* dKO	*Mpz*-cre^ERT2^ tamoxifen-inducible	*Hdac1*^flox/flox^ (MGI:4440556) *Hdac2*^flox/flox^ (MGI:4440560)	Early entry into SC repair phenotype Impaired remyelination Improved regeneration	[47]
*Hdac1/2* dKO	*Dhh*-cre	*Hdac1*^flox/flox^ (MGI:4440556) *Hdac2*^flox/flox^ (MGI:4440560)	Increased eEF1A1 acetylation and low *Sox10* levels	[48]
*Hdac3* cKO	*Dhh*-cre	*Hdac3*^flox/flox^ (MGI:3831693)	Hypermyelination	[5]
*Hdac3* cKO	*Cnp*-cre (MGI:3051635)	*Hdac3*^flox/flox^ (MGI:3831693)	Hypermyelination during SC development and progressive demyelination at adult stage in peripheral nerves	[5]
*Hdac3* iKO	*Plp1*-cre^ERT^ tamoxifen-inducible (MGI:2450391)	*Hdac3*^flox/flox^ (MGI:3831693)	Accelerated SC maturation Improved Remyelination Improved functional recovery	[5]
*Hdac3* cKO	*Mpz*-cre (MGI:2450448)	*Hdac3*^flox/flox^ (MGI:4433408)	Myelin homeostasis	[49]
*Hdac4* cKO	*Mpz*-cre	*Hdac4*^flox/flox^ (MGI:4418117)	Impaired Myelination Impaired Remyelination	[4,6]
*Hdac5 null KO*	-	*Hdac5*^−/−^ (MGI:3056065)	No effect	[4,6]
*Hdac4/5 dKO*	*Mpz*-cre	*Hdac4* ^flox/flox^ *Hdac5* ^−/−^	Delayed myelination Impaired remyelination	[4,6]
*Hdac7 cKO*	*Mpz*-cre	*Hdac7*^flox/flox^ (MGI:3693628)	No effect	[6]
*Hdac4/5/7 tKO*	*Mpz*-cre	*Hdac4* ^flox/flox^ *Hdac5* ^−/−^ *Hdac7* ^flox/flox^	Delayed Myelination Impaired Remyelination Altered Remak bundles	[6]

**Table 2 ijms-23-02996-t002:** HDACs inhibitor or activator used in in vivo Schwann cell (SC) or peripheral nervous system (PNS) studies. (Note: Mocetinostat is also an HDAC3 and HDAC11 inhibitor. Theophylline is also an HDAC1 enhancer).

**HDAC Inhibitor**		**Effects on Schwann Cell Biology**	**Paper**
TSA (Trichostatin A)	Class I and II Inhibitor	Lower SC proliferation Low LAMP-1 and p75 NTR levels Axonal degradation inhibition	[Kim 2019]
Mocetinostat	HDAC1/2 specific inhibitor	Increased Sox10 levels in neural crest explants	[Jacob 2014]
		Improved sensory functions Improved regeneration	[Brügger 2017]
PDA106 (Pimelic diphenylamide)	HDAC3	Improved remyelination	[He 2018]
PBA (Sodium phenylbutyrate)	Class I and class II a Inhibitor	Reduced pro-inflammatory cytokines Improved axonal regrowth and remyelination	[Yadav 2021]
VPA (Valproic acid)	Class I Inhibitor	Functional sciatic nerve recovery Improved regeneration Increased SCs proliferation	[Wu 2021]
**HDAC Activator**		**Effects on Schwann Cell Biology**	**Paper**
Theophylline	HDAC2 activator	Increased Sox10 levels and Improved remyelination	[Duman 2020]

**Table 3 ijms-23-02996-t003:** Examples of HDACs inhibitors in clinical trials of diseases of CNS and PNS (from https://clinicaltrials.gov/) (Accessed on 18 February 2022).

Compound	Disease	Phase	Status
Vorinostat (pan inhibitor)	Alzheimer’s disease	Phase 1	Recruiting
CKD-504	Huntington disease	Unknown	Unknown
Nicotinamide (HDACi3)	Friedreich’s ataxia	Phase 0	Not recruiting yet
Vorinostat (pan inhibitor)	Niemann–Pick Disease	Phase 1/2	Completed
Ricolinostat (hDACi6)	Painful diabetic neuropathy	Phase 2	Recruiting
AR-42 (OSU-HDAC42)	Vestibular Schwannoma Neurofibromatosis Type 2	Early Phase 1	Active, not recruiting
REC-2228 (OSU-HDAC42)	Neurofibromatosis Type 2	Phase 2/3	Not yet recruiting

## Data Availability

Not applicable.

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
