# Peer review of "Emerging Role of HDACs in Regeneration and Ageing in the Peripheral Nervous System: Repair Schwann Cells as Pivotal Targets"

_ijms, 2022, doi:10.3390/ijms23062996_

Round 1
Reviewer 1 Report
This review comprehensively covers the role of HDACs in the PNS and in Schwann cells. The review is extremely well written, very informative and a pleasure to read. It covers the fields from the discovery of HDACs in Schwann cells health, aging and diseases to the most modern therapies to modulate HDACs, with a perspective of using this knowledge in peripheral neuropathies. Overall, a very interesting read that cover perfectly the current state of the art.
One minor comment, it would have been also interesting to have a perspective on the substrate of HDACs – Acetyl and Acetyl-CoA, which connect to energetic metabolism and amplify even more the indirect role that HDACs could play based on the neuroglia interaction.
Reviewer 2 Report
In this review, the authors elaborate on the emerging role of HDACs in regeneration and aging in the PNS and outline repair Schwann cells as pivotal targets. The manuscript is very well written and provides a useful overview about this interesting topic.
There are some issues that should be addressed before acceptance:
- It is evident that the authors published significant articles essential for the field, however, they should also cite other studies that contributed to the current knowledge. For example, the paragraph about the repair Schwann cell (lines 169-190) should include citations of early key studies on the importance of transcription factor JUN in PNS injury e.g. https://www.ncbi.nlm.nih.gov/pmc/articles/PMC6577506/ and the Schwann cell capacity of myelin clearance e.g. https://pubmed.ncbi.nlm.nih.gov/12112437/ , and first studies on human repair Schwann cells e.g. https://pubmed.ncbi.nlm.nih.gov/27545331/.
- The Figures showing hdac expression changes during development and injury, in aged and young mice are derived from the Sciatic Nerve Atlas (Gerber et al 2021) determined by bulk RNA sequencing. As there are now several single cell sequencing studies about rodent peripheral nerves available (e.g. https://www.pnas.org/content/117/17/9466 , https://www.ncbi.nlm.nih.gov/pmc/articles/PMC8019921/ , https://www.nature.com/articles/s41593-021-01005-1 ), the review would significantly benefit if the authors would explore these data sets and provide an up to date overview for hdac expression changes in the Schwann cell subpopulations in intact and injured peripheral nerves.
Minor points:
- It is pointed out in lines 304-305, 307-308 and 310-311 that Schwann cells are not the main cell source of Hdacs expression during development. Although it is appreciated that the authors stress this point, these sentences are very similar and should be revised.
- As the authors use the data set of Gerber et al 2021, a short description about it would be appropriate (how many nerves have been analyzed, are the shown expression changes significant, etc)
- Line 401: a ‘point’ is missing at the end of the sentence.
Reviewer 3 Report
Gomez-Sanchez and colleagues summarize the main findings on the role of HDACs on SC biology and response to injury, and in ageing. Moreover, authors report on the use of novel HDAC inhibitors in a therapeutic perspective to improve nerve regeneration.
HDAC genetic compensation in SC is also discussed.
The review is complete and well written. Nevertheless, authors are asked to comment on the following general points.
- As nerve regeneration is orchestrated by a plethora of signals from many different cell types, and HDACs are mostly expressed by cells other that SCs within the nerve, this Reviewer wonders whether conditional/inducible SC KO mouse models are the most suitable tools to dissect HDACs’ role in nerve regeneration.
-Table 1. The effects of conditional/inducible KO of HDACs in SC are sometimes controversial. For example, in the Hdac1/2 dKO (ref. 52) remyelination is impaired but regeneration is improved. As remyelination is a conditio sine qua non for restoration of motor function, how these 2 apparently opposite results can be interpreted?
-This Reviewer is sometimes confused by controversial results reported in paragraph. 3.4. The role of HDACs in repair Schwann Cells and remyelination. Here the inhibition of class I and II HDACs by TSA is reported to interfere with SC conversion into repair SC, namely with nerve regeneration, while another class I inhibitor, VPA, improves the same process.
-Moreover, why should TSA delay nerve Degeneration?
- 3.6. Hdac levels after injury and ageing. The trend of expression of HDACs after an acute injury, which is more likely to repair, shows an upregulation of a number of HDAC subtypes followed by their return to basal levels. This suggests the importance of a precise modulation of HDACs expression that should be taken into account in chronic neurodegenerative contexts in a therapeutic perspective. Please comment on this point.
Minor point:
- 2. Class II HDACs. Please add the important contributions of class I, II HDACs HDAC3 and HDAC5 in axonal regeneration by citing the following studies:
Hervera, A et al. PP4-dependent HDAC3 dephosphorylation discriminates between axonal regeneration and regenerative failure. EMBO J. 2019, 38, e101032.
Cho, Y. and Cavalli, V. HDAC5 is a novel injury-regulated tubulin deacetylase controlling axon regeneration. EMBO J. 2012, 31, 3063–3078.
